# *Ishige okamurae* Extract Suppresses Obesity and Hepatic Steatosis in High Fat Diet-Induced Obese Mice

**DOI:** 10.3390/nu10111802

**Published:** 2018-11-20

**Authors:** Young-Jin Seo, Kippeum Lee, Ji-Hyeon Song, Sungwoo Chei, Boo-Yong Lee

**Affiliations:** Department of Food Science and Biotechnology, College of Life Science, CHA University, Seongnam 13488, Kyeonggi, Korea; youngjinseo92@gmail.com (Y.-J.S.); joy4917@hanmail.net (K.L.); redcross0313@naver.com (J.-H.S.); sungwoochei@gmail.com (S.C.)

**Keywords:** *Ishige okamurae*, obesity, hepatic steatosis, 3T3-L1 cells, high fat diet, mice

## Abstract

Obesity is caused by the expansion of white adipose tissue (WAT), which stores excess triacylglycerol (TG), this can lead to disorders including type 2 diabetes, atherosclerosis, metabolic diseases. *Ishige okamurae* extract (IOE) is prepared from a brown alga and has anti-oxidative properties. We investigated the detailed mechanisms of the anti-obesity activity of IOE. Treatment with IOE blocked lipid accumulation by reducing expression of key adipogenic transcription factors, such as CCAAT/enhancer-binding protein alpha (C/EBPα) and peroxisome proliferator-activated receptor gamma (PPARγ), in 3T3-L1 cells. Administration of IOE to high fat diet (HFD)-fed mice inhibited body and WAT mass gain, attenuated fasting hyperglycemia and dyslipidemia. The obesity suppression was associated with reductions in expression of adipogenic proteins, such as C/EBPα and PPARγ, increases in expression of lipolytic enzymes, such as adipose triglyceride lipase (ATGL) and hormone-sensitive lipase (HSL), in WAT of HFD-fed mice. In addition, IOE-treated mice had lower hepatic TG content, associated with lower protein expression of lipogenic genes, such as diglyceride acyltransferase 1 (DGAT1), sterol regulatory element-binding protein 1 (SREBP1), fatty acid synthase (FAS). IOE treatment also reduced serum free fatty acid concentration, probably through the upregulation of β-oxidation genes, suggested by increases in AMPKα and CPT1 expression in WAT and liver. In summary, IOE ameliorates HFD-induced obesity and its related metabolic disease, hepatic steatosis, by regulating multiple pathways.

## 1. Introduction

Obesity is associated with insulin resistance and dyslipidemia, which can lead to chronic disease in adults, is increasing in prevalence worldwide. The prevention and treatment of obesity is essential for a healthy life. The main cause of obesity is the hypertrophy of white adipose tissue (WAT) caused by energy imbalance [1]. WAT stores lipids in the form of triacylglycerol (TG) to regulate energy balance [2]. However, consumption of food containing more energy than is expended causes adipose tissue expansion and ultimately obesity [3]. The TG stored in adipose tissue can be hydrolyzed to release glycerol and free fatty acids (FFA), which are used as fuel molecules in liver, muscle, and other tissues [4]. However, adipocytes also secrete adipokines, including hormones, cytokines and other proteins that have specific biological functions [5,6]. For these reasons, adipose tissue has an important impact on physiological processes such as adipocyte development and energy homeostasis.

Adipogenesis is a complex process involving a series of changes in cell morphology and gene expression [7]. Preadipocytes of the 3T3-L1 cell line are differentiated into mature adipocytes by the synergistic effect of the adipogenic factors CCAAT/enhancer-binding protein alpha (C/EBPα) and peroxisome proliferator-activated receptor gamma (PPARγ), which are induced by treatment with a hormone cocktail [8,9]. These adipogenic transcription factors induce the expression of genes involved in TG accumulation, such that the cells synthesize and accumulate lipids [10]. The amount of TG that accumulates in adipocytes is determined by the relative rates of lipolysis and lipid synthesis, which both take place continuously.

Lipolysis is the degradation of TG to yield FFA and glycerol, can be induced by AMP-activated protein kinase alpha (AMPKα) in adipocytes [11]. AMPKα acts as an energy sensor and is activated by an increase in the intracellular AMP/ATP ratio [12]. Activation of AMPKα promotes ATP production by increasing the rate of oxidation of glucose or fatty acid and decreases ATP consumption [13]. The activity of adipose triglyceride lipase (ATGL), which hydrolyzes TG into FFA and diacylglycerol, is also promoted by AMPKα [14]. Then, the phosphorylation of hormone-sensitive lipase (HSL) leads to the hydrolysis of diacylglycerol to liberate FFA and monoacylglycerol [15]. The FFA produced by lipolysis migrate into the circulation, they can accumulate here or in other tissues [4]. The excess FFA and other lipids associated with ingestion of a high fat diet (HFD) accumulate in the blood, causing insulin resistance [16,17]. Under these circumstances, expression of lipolysis-induced ATGL and HSL is inhibited to limit the accumulation of FFA in the circulation [18].

The FFA produced by lipolysis are transported to the liver in the circulation and are used to synthesize TG by diglyceride acyltransferase 1 (DGAT1) [19]. In the liver, TG accumulation occurs following upregulation of proteins involved in de novo lipogenesis, such as sterol regulatory element-binding protein 1 (SREBP1) and fatty acid synthase (FAS) [20]. When hepatic lipid accumulates chronically, nonalcoholic fatty liver disease (NAFLD) develops [21]. This obesity-related NAFLD can then progress to nonalcoholic steatohepatitis (NASH), which is further characterized by chronic inflammation and fibrosis [22]. Conversely, hepatic lipid accumulation is reduced by the upregulation of genes involved in FFA oxidation, such as mitochondrial carnitine palmitoyltransferase I (CPT1) [23,24]. This β-oxidation also provides energy in the form of ATP. The activity of AMPKα reduces fat accumulation by inducing lipolysis and inducing β-oxidation in adipose tissue and liver.

A number of studies show that seaweed extracts can ameliorate components of the metabolic syndrome, such as obesity and insulin resistance [25,26,27]. *Ishige okamurae* extract (IOE) is obtained from a brown algal seaweed and is reported to have anti-inflammatory effects [28]. Furthermore, diphlorethohydroxycarmalol isolated from *Ishige okamurae* has been shown to have anti-oxidant activity, protecting against cell damage in mice [29,30]. In addition, phlorotannins, polyphenolic components of the physiologically active extract of *Ishige okamurae*, have anti-oxidant and neuroprotective properties [31,32]. Moreover, polysaccharides such as fucoidan, which are also present in brown algae, are reported to reduce fat accumulation in HFD-induced obese mice [33]. Here, we investigated the mechanism of the anti-obesity effect of IOE. We show that IOE reduces HFD-induced obesity in mice, likely by inhibiting lipid synthesis and activating lipolysis and fatty acid oxidation proteins. Furthermore, IOE ameliorates fatty liver, likely by regulating lipid synthesis and β-oxidation-related factors in this organ. Thus, consumption of IOE may be effective at ameliorating obesity and its associated metabolic diseases.

## 2. Materials and Methods

### 2.1. Experimental Materials

Cell reagents, including Dulbecco’s modified Eagle’s medium (DMEM), bovine calf serum (BCS), fetal bovine serum (FBS), penicillin-streptomycin (P/S), insulin, trypsin/EDTA, were purchased from Gibco (Gaithersburg, MD, USA). Dexamethasone, 3-isobutyl-1-methylxanthine (IBMX), isopropanol, Oil Red O, chemical reagents were mainly purchased from Sigma-Aldrich (St Louis, MO, USA). IOE powder (Lot No. SW8D10SA) was acquired from Jeju National University (Jeju, Korea). IOE was prepared using 50% ethanol for 24 h at room temperature. After 50% ethanol extraction, it was concentrated by vacuum evaporation at 40 °C, filtered, and then it was powdered by using spray drying. The component analyses of IOE were conducted on a high performance liquid chromatography (HPLC) system (Agilent Technologies, Palo Alta, CA, USA) equipped with an Agilent poroshell 120 EC-C18 column (4.6 × 150 mm, 4 μm) and a UV detector (230 nm). The mobile phase consisted of solvent A (0.1% formic acid in water) and solvent B Acetonitrile (ACN) containing 0.1% formic acid). The conditions of elution were as follows: 5–40% B for 40 min to 100% B for 10 min, followed by 10 min, re-equilibration time of the column. The flow rate was maintained at 0.3 mL/min and the injection volume was 10 μL. Index component of IOE, Ishophloroglucin A of retention time was detected at 35.4 min and content of Ishophloroglucin A in IOE was 61.52 μg/mL ± 20% (1.24% ± 20%) [34]. Total polyphenol content of IOE was 4.2%. Ishophloroglucin A, index component of IOE, contains 29.52% in total polyphenol content of IOE.

### 2.2. Cell Culture

Mouse 3T3-L1 preadipocytes were purchased from the ATCC (Manassas, VA, USA). 3T3-L1 adipose precursor cells were cultured on 60 mm plates in DMEM containing 10% BCS for cell culture until confluent. To induce differentiation, these cells were incubated in differentiation medium containing 10% FBS and MDI (4 μg/mL insulin, 10 μM dexamethasone, and 0.5 mM IBMX). After 2 days, the medium was replaced with medium containing 10% FBS and 4 μg/mL insulin every 2 days to promote the differentiation.

### 2.3. Cell Viability Assay

To determine the appropriate concentration of IOE to be used, an XTT (2,3-Bis-(2-methoxy 4-nitro-5-sulfophenyl)-2*H*-tetrazolium5-carboxanilide salt) cell viability assay was performed to assess cytotoxicity. For this purpose, 3 × 10^4^ 3T3-L1 preadipocytes/well were cultured in 96-well plates for 24 h, then treated with 0–100 μg/mL IOE. After 24 h, 40 µL XTT solution containing XTT and PMS reagent (Welgene, Deagu, Korea) were added to each well and the cells were further incubated for 4 h in a 5% CO_2_ incubator. Absorbance was then measured at 450 and 690 nm using an ELISA plate reader (BioTek, Winooski, VT, USA).

### 2.4. Oil Red O Staining

To assess the cellular lipid content, Oil Red O staining was performed during differentiation. The 3T3-L1 adipocytes were fixed with 4% formaldehyde in a CO_2_ incubator for 5 min and at room temperature for 1 h. The cells were then washed with 60% isopropanol, Oil Red O solution was added, and they were incubated at room temperature for 20 min. The Oil Red O solution was then removed, the cells were washed three times with PBS, and the stained cells were assessed by scanner. For quantitative analysis, the intracellular lipid and Oil Red O were solubilized with 100% isopropanol and 100 µL from each well was transferred to a 96-well plate. The absorbance of each aliquot was then measured at 490 nm using an ELISA plate reader.

### 2.5. Animal Studies

All animal studies were approved by the Institutional Animal Care and Use Committee (IACUC) of CHA University (Approval Number 180003). Male ICR mice were purchased from Japan SLC Inc. (Shizuoka, Japan) at 4 weeks of age. Mice were kept in temperature and humidity-modulated facilities on a 12 h light/dark cycle. After a 1 week period of adaptation, mice were fed a standard diet (SD) purchased from Envigo (Huntingdon, Cambridgeshire, UK), a HFD (Joongah Bio, Suwon, Korea), or a HFD supplemented with IOE 100 and 300 mg/kg/day for 6 weeks. The HFD contained 20 kcal% carbohydrate, 20 kcal% protein, and 60 kcal% fat. The SD and HFD groups were administered with vehicle in place of IOE. During the experiment, body mass, fasting blood glucose, and dietary intake were measured weekly. Subsequently, after fasting overnight, the mice were terminally anesthetized using CO_2_ and tissues samples were collected by dissection.

### 2.6. Blood Parameter Analysis

After 6 weeks, blood was collected by cardiac puncture immediately after euthanasia and serum was separated by clotting for 30 min at 4 °C and centrifugation at 2000× *g* for 20 min. The serum concentrations of TG, FFA, cholesterol, aspartate transaminase (AST), alanine transaminase (ALT), and lactate dehydrogenase (LDH) activities were determined using colorimetric assay kits (Roche, Basel, Switzerland). Serum cholesterol, TG, FFA, AST, ALT, and LDH were measured using Cobas 8000 c702 Chemistry Analyzer (Roche, Basel, Switzerland).

### 2.7. Histological Analysis

WAT and liver biopsies were fixed with 4% paraformaldehyde and embedded in paraffin. Sections were obtained, stained with hematoxylin and eosin (H&E), and analyzed by light microscopy. Liver triglyceride content was assayed using a commercial TG assay kit (Cayman Chemical Company, Ann Arbor, MI, USA).

### 2.8. Western Blot Analysis

Cells and tissues were lysed in lysis buffer for 30 min, and protein quantification was performed using the Bradford assay (Bio-Rad Laboratories, Hercules, CA, USA). Lysates containing 20 µg protein were electrophoresed on 8–12% SDS-PAGE gels and electrotransferred to membranes for ~2 h. The membranes were then immersed in 5% skim milk blocking buffer for 1 h and incubated with a primary antibody at 4 °C overnight. After several washes, secondary antibody and chemiluminescent detection reagents were sequentially applied. Antibodies targeting C/EBPα, PPARγ, p-AMPKα, AMPKα, CPT1, SREBP1, FAS, DGAT1, and glyceraldehyde-3-phosphate dehydrogenase (GAPDH) were purchased from Santa Cruz Biotechnology (Dallas, TX, USA), and ATGL, p-HSL, and HSL antibodies were from Cell Signaling Technology (Beverly, MA, USA).

### 2.9. Statistical Analysis

The results are presented as means ± standard deviations (S.D). Comparisons between experimental groups were made using one-way analysis of variance (ANOVA) followed by Tukey’s test (SPSS 12.0 software, Chicago, IL, USA). Statistical differences among the groups are represented using a, b, c, and d. *p*-values with different letters are significantly different, *p* < 0.05.

## 3. Results

### 3.1. IOE Reduces Lipid Accumulation in 3T3-L1 Adipocytes

To investigate whether IOE reduces adipocyte differentiation, we first assessed IOE for toxic effects using an XTT assay. As shown in Figure 1A, we found that IOE was not cytotoxic in the range of 0–25 μg/mL. We therefore decided to use concentrations of 6.25, 12.5, or 25 μg/mL IOE during adipocyte differentiation.

To assess the inhibitory effect of IOE on adipogenesis, we differentiated 3T3-L1 preadipocytes into adipocytes with MDI differentiation medium in the presence or absence of IOE for 8 days. As shown in Figure 1B, we found that IOE reduced the number of lipid droplets in a dose-dependent manner using Oil Red O staining. As shown in Figure 1C, IOE also suppressed the expression of adipogenic transcription factors (C/EBPα and PPARγ) in a dose-dependent manner by western blot analysis. During adipocyte differentiation, a high concentration of IOE reduced adipogenesis to the extent that the cells appeared like undifferentiated controls. Thus, IOE treatment caused significantly lower lipid accumulation than in differentiated control cells, probably through repression of adipogenic proteins.

### 3.2. IOE Reduces Adiposity in HFD-Induced Obese Mice

To confirm an anti-obesity effect of IOE in vivo, we fed mice a HFD ± IOE for 6 weeks. Weekly measurements of body mass during this period are shown in Figure 2A. HFD-fed mice and those that were also administered IOE for 6 weeks gained 26.8 g and 18 g, respectively; that is, the IOE-treated mice gained less weight than HFD-fed mice. As shown in Figure 2C, IOE-treated mice gained substantially less subcutaneous and visceral WAT mass than HFD-fed controls. In addition, brown adipose tissue (BAT) mass tended to increase in IOE-treated mice than solely HFD-fed mice. However, there was no statistical significance between the treated and untreated groups. As shown in Figure 2B, food intake was not statistically different. Finally, other tissue masses were not affected by HFD or IOE treatment (Table 1). Thus, IOE does not cause weight loss due to changes in dietary intake, but by reductions in adipose tissue mass.

### 3.3. IOE Ameliorates Deleterious Changes in Blood Metabolic Parameters in Hfd-Induced Obese Mice

Obesity, especially visceral obesity, is associated with high serum cholesterol concentrations [35]. This hypercholesterolemia is a frequent component of the dyslipidemia that is associated with insulin resistance and its associated metabolic diseases [36]. To determine whether IOE treatment can ameliorate obesity-related abnormalities in blood parameters, we analyzed circulating total cholesterol, low-density lipoprotein (LDL)-cholesterol and high-density lipoprotein (HDL)-cholesterol concentrations. As shown in Table 2 and Table 3, IOE-treated mice had lower total cholesterol, LDL-cholesterol, and blood glucose levels, but higher HDL-cholesterol levels, than solely HFD-fed mice. In addition, IOE treatment was associated with lower activities of hepatic metabolic enzymes (ALT, AST and LDH), which are indicative of liver damage [21], than HFD feeding alone.

### 3.4. Treatment with IOE Is Associated with Smaller Lipid Droplets and Lower Expression of Adipogenic Transcription Factors in the WAT of HFD-Induced Obese Mice

The fat droplets in visceral WAT were larger in HFD-fed mice than in mice on a normal diet [37]. As shown in Figure 3A, mice administered with IOE for 6 weeks had smaller visceral adipocytes than mice fed a HFD alone. To determine the effect of IOE on adipogenesis, we measured the expression of the key adipogenic proteins C/EBPα and PPARγ by western blotting, as shown in Figure 3B, HFD-fed mice had higher expression of these proteins than SD-fed mice. However, IOE-treated mice showed lower expression of C/EBPα and PPARγ in WAT. These data suggest that IOE suppresses the HFD-induced increase in lipid droplet size by reducing the expression of key adipogenic transcription factors in WAT.

### 3.5. IOE Increases the Expression of AMPKα and Lipolytic Enzymes in 3T3-L1 Adipocytes and the WAT of HFD-Induced Obese Mice

To determine whether other mechanisms might also be involved in the IOE-induced reduction in lipid accumulation, we measured the expression of lipolytic enzymes. Several studies show that AMPKα phosphorylation results in reductions in the expression of adipogenic proteins and increases in expression of lipolytic enzymes [11,38]. As shown in Figure 4A, IOE-treated mice demonstrated greater phosphorylation of AMPKα, which was lower in HFD-fed than SD-fed mice. HFD-fed mice showed lower WAT expression of the lipolytic enzymes ATGL and HSL than SD-fed mice (Figure 4B), but IOE-treated mice showed higher levels of expression than HFD-fed mice. Similarly, IOE treatment induced the phosphorylation of AMPKα and the expression of ATGL and HSL in 3T3-L1 cells (Figure 4C). Thus, IOE may also reduce TG storage by inducing lipolysis in 3T3-L1 cells and HFD-induced obese mice.

### 3.6. IOE Induces FA Oxidation in 3T3-L1 Adipocytes and the WAT of HFD-Induced Obese Mice

High levels of FFA are a component of dyslipidemia and can cause insulin resistance [16,39]. Induction of CPT1 expression is required for β-oxidation and protects against the insulin resistance induced by fatty acids [40]. As shown in Figure 5A, HFD-fed mice expressed less CPT1 than SD-fed mice, but IOE administration was associated with higher CPT1 expression in the WAT of HFD-fed mice. As shown in Figure 5B, IOE treatment also stimulated higher CPT1 expression in 3T3-L1 cells. Thus, IOE promoted the oxidation of FFA by upregulating CPT1 expression, thereby reducing the circulating concentrations of TG and FFA in HFD-fed mice (Figure 5C,D).

### 3.7. Treatment with IOE Reduces Hepatic Lipid Accumulation in HFD-Induced Obese Mice and Regulates Lipogenesis and β-Oxidation

HFD-induced visceral obesity predisposes towards metabolic abnormalities such as NAFLD [41]. We assessed the hepatic lipid content of the mice using H&E staining and found that the staining intensity was greater in HFD-fed than in SD-fed mouse liver (Figure 6A), whereas IOE-treated mice showed lower staining intensity. Consistent with this, the HFD-fed mice had a higher TG content than the SD-fed mice (Figure 6B), whereas the TG content was lower in IOE-treated than in untreated HFD-treated mice.

To determine how IOE affects hepatic lipid turnover, we assessed de novo lipogenesis, TG synthesis, and β-oxidation by western blotting of key proteins. FFA are delivered to the liver via the portal vein and are used to synthesize TG, which involves the upregulation of DGAT1 expression [19]. As shown in Figure 6C, IOE-treated mice showed lower hepatic DGAT1 expression than HFD-fed control mice. In addition, hepatic TG accumulation occurs secondary to the upregulation of de novo lipogenesis, mediated by molecules such as SREBP1 and FAS [42]. IOE-treated mice demonstrated lower expression of these lipogenic genes than control HFD-fed mice (Figure 6C). AMPKα is also reported to not only reduce the expression of transcription factors required for lipogenesis, such as SREBP-1c, but also to induce fat oxidation, thereby reducing lipid accumulation in the liver [42,43]. As shown in Figure 6D, HFD-fed mice demonstrated lower AMPKα phosphorylation than SD-fed mice, but AMPKα expression was higher in IOE-treated mice than HFD-fed controls. Thus, IOE intake increases the expression of the β-oxidative factor CPT1, which may be induced by AMPKα activation (Figure 6D), resulting in FFA oxidation. To summarize, we have presented evidence that IOE inhibits TG synthesis and upregulates β-oxidation to prevent hepatic TG accumulation.

## 4. Discussion

Obesity predisposes towards the pathological abnormalities observed in the metabolic syndrome [41]. In addition, the prevalence of diet-induced NAFLD has been increasing worldwide [44]. The resolution of obesity and NAFLD can involve reducing the size of adipose tissue depots, decreasing FFA uptake and de novo lipogenesis in the liver, and/or upregulating the oxidation of lipids in adipose tissue and liver.

IOE contained 4.2% of polyphenols and Ishophloroglucin A, an indicator component of IOE, was also composed of polyphenol. Polyphenol-rich compounds are well known to have anti-obesity and blood glucose control activity [45]. It has been reported that polyphenol-rich compound inhibits body weight, hyperglycemia, and hyperinsulinemia via enhancement of energy expenditure and fat oxidation [45,46]. Previous studies have shown that diphlorethohydroxycarmalol isolated from *Ishige okamurae* reduces blood sugar [47]. We anticipated that IOE might have anti-obesity properties. The results of this investigation demonstrate that IOE reduces adipose tissue mass, total body mass, and hepatic lipid content.

The differentiation of adipose tissue involves the upregulation of key lipogenic proteins such as C/EBPα and PPARγ [10], which promote energy storage in the form of TG. We have shown that IOE inhibits the differentiation of preadipocytes, probably by reducing the expression of C/EBPα and PPARγ expression. In addition, TG stored in visceral fat can be hydrolyzed to liberate glycerol and FFA by lipolytic enzymes such as ATGL and HSL when necessary [48], and we have shown that IOE promotes the expression of these enzymes in WAT, which would tend to reduce adipocyte size. The FFA generated is released into the circulation and transported to the liver or muscle, both of which utilize a great deal of FFA [4]. However, uptake of excessive amounts of FFA by these tissues causes insulin resistance and cardiovascular disease [16]. One strategy to prevent this is to encourage the β-oxidation of the excess FFA, which is an important means of energy production.

AMPKα is a crucial upstream regulator of lipolysis and lipid oxidation in WAT and liver [49]. CPT1 is an enzyme that is required for the transport of FFA into mitochondria [50], after which fatty acyl CoA is oxidized to generate acetyl CoA, which is then processed by the tricarboxylic acid (TCA) cycle and electron transport chain to generate ATP [51]. HFD-induced obesity is associated with lower rates of β-oxidation, meaning that less FFA is consumed. IOE may induce greater use of FFA as an energy source by upregulating AMPKα and CPT1 expression in 3T3-L1 cells and HFD-induced obese mice. This would have the effect of reducing the amount of FFA released into the circulation and the degree of adiposity.

HFD-induced obesity is also associated with the presence of higher concentrations of cholesterol, blood glucose, TG, and FFA in the circulation, and higher concentrations of markers of liver damage, such as ALT, AST, and LDH [52]. IOE ameliorated the hyperglycemia and dyslipidemia present in the HFD-fed mice, indicated by lower circulating concentrations of cholesterol, glucose, TG, and FFA. High FFA concentrations and chronic inflammation in WAT are considered to be two key factors contributing to the progression of liver pathology in NAFLD [21]. Hepatic lipid accumulation is caused by greater transfer of FFA from adipose tissue and de novo lipogenesis [53]. The resulting continual lipid accumulation in the liver can lead to the progression of NAFLD from hepatic steatosis to NASH. We have shown that IOE ameliorates hepatic TG accumulation, probably by inhibiting the uptake of excess FFA. DGAT1, a major enzyme involved in TG synthesis, plays an important role in fatty liver development [19]. Furthermore, de novo lipogenesis is mediated by SREBP1 and its downstream target, FAS [54]. IOE reduced DGAT1 expression, which is necessary for TG synthesis from diacylglycerol, as well that of SREBP1 and FAS. In addition, IOE administration induced the expression of AMPKα and CPT1, which promote or are necessary for lipid oxidation, which would facilitate the use of FFA to generate energy. Therefore, IOE inhibits hepatic steatosis by reducing TG accumulation in the liver. In this way, IOE could not only ameliorate obesity, but also obesity-associated metabolic diseases, such as dyslipidemia and hepatic steatosis.

## 5. Conclusions

IOE treatment reduced lipid droplet size in 3T3-L1 cells, probably by inhibiting expression of adipogenic factors, such as C/EBPα and PPARγ. IOE treatment reduced body weight gain by preventing an increase in WAT mass and ameliorated hyperglycemia and hypercholesterolemia in HFD-fed mice. The lower adiposity was likely mediated through a reduction in expression of adipogenic proteins such as C/EBPα and PPARγ and the induction of lipolytic enzymes such as ATGL and HSL in WAT. IOE treatment also prevented fatty liver, probably by inhibiting hepatic lipogenesis, indicated by lower hepatic expression of DGAT1, SREBP1, and FAS. Serum TG and FFA levels were also lower, likely as a result of IOE treatment upregulating proteins involved in β-oxidation genes, such as AMPKα and CPT1. Thus, IOE ameliorates HFD-induced obesity and NAFLD by regulating a number of pathways.

## Figures and Tables

**Figure 1 nutrients-10-01802-f001:**
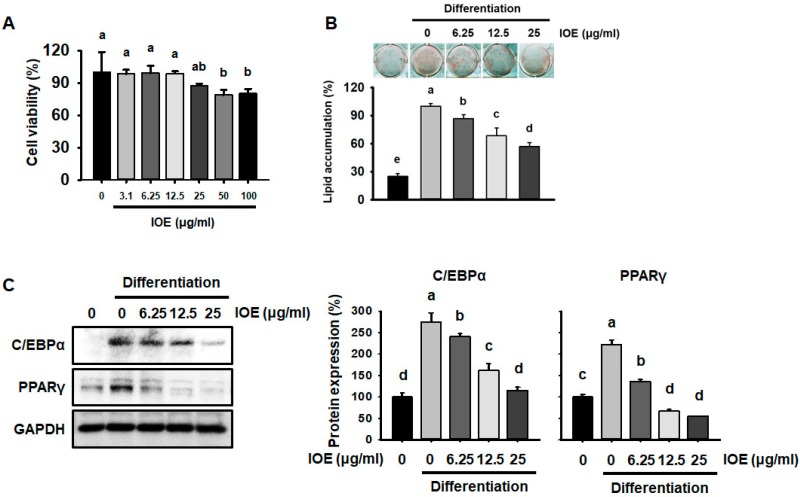
IOE reduces lipid accumulation in 3T3-L1 adipocytes. (**A**) Cell viability of 3T3-L1 preadipocytes treated with IOE, determined over 24 h using an XTT assay. (**B**) Effect of IOE on lipid accumulation, determined using Oil Red O staining, in 3T3-L1 adipocytes. A differentiation-inducing cocktail, with or without IOE, was added to 3T3-L1 adipocytes for 8 days. (**C**) Protein expression of adipogenic transcription factors (C/EBPα and PPARγ) after 8 days of incubation of 3T3-L1 adipocytes in differentiation medium. Values with different letters are significantly different; *p* < 0.05 (a > b > c > d).

**Figure 2 nutrients-10-01802-f002:**
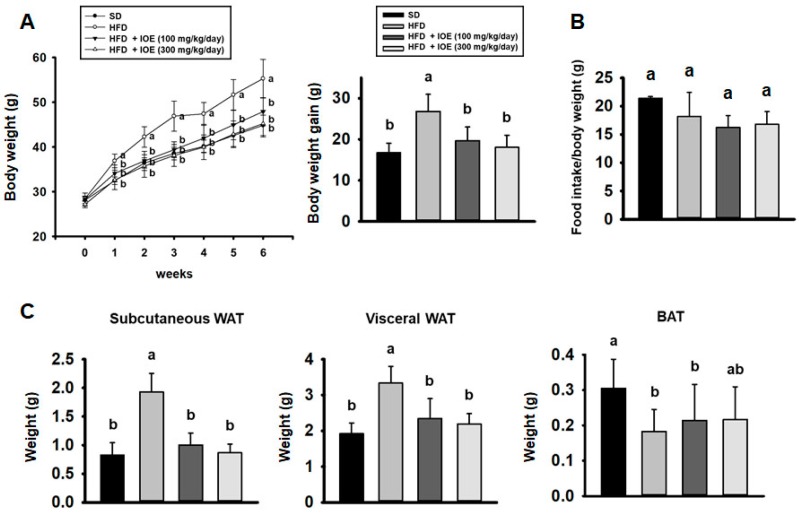
IOE reduces the adiposity of HFD-induced obese mice. Body mass gain (**A**), and food intake per unit body mass (**B**), measured during 6 weeks’ treatment with or without IOE. (**C**) Subcutaneous and visceral WAT mass. Data are expressed as mean ± S.D (*n* = 6). Values with different letters are significantly different; *p* < 0.05 (a > b).

**Figure 3 nutrients-10-01802-f003:**
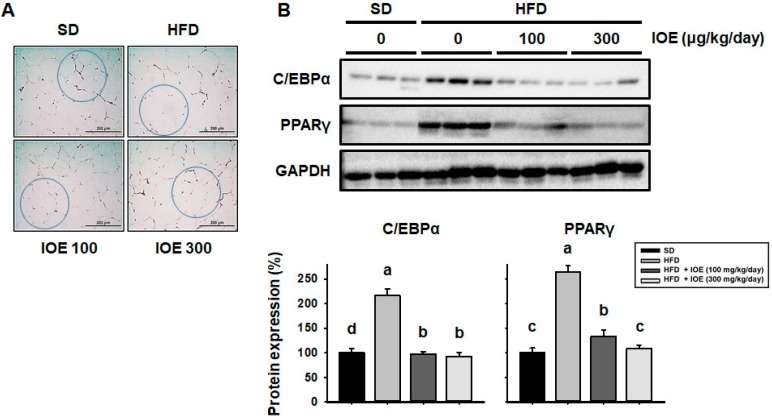
IOE treatment is associated with smaller lipid droplets in WAT and lower expression of key lipogenic transcription factors in the WAT of HFD-induced obese mice. (**A**) WAT morphology was analyzed using hematoxylin and eosin (H&E) staining. (**B**) Western blotting of lipogenic transcription factors (C/EBPα and PPARγ) in WAT. Data are expressed as mean ± S.D (*n* = 6). Values with different letters are significantly different; *p* < 0.05 (a > b > c).

**Figure 4 nutrients-10-01802-f004:**
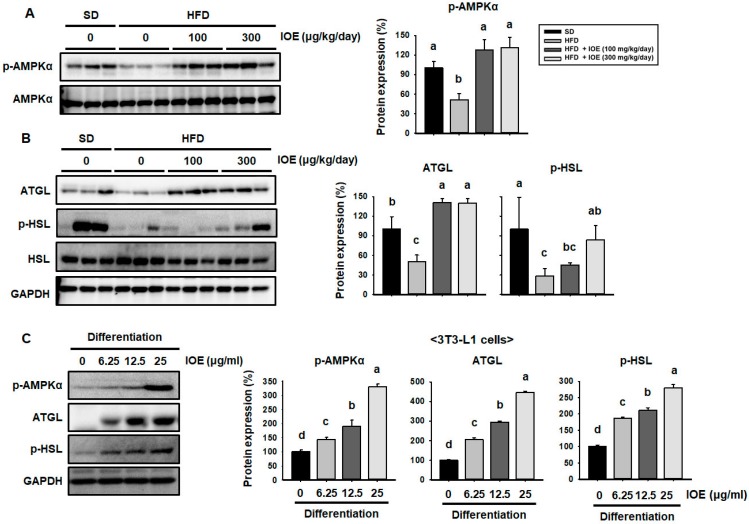
IOE treatment increases phosphorylation of AMPKα and the expression of lipolytic enzymes in 3T3-L1 adipocytes and the WAT of HFD-induced obese mice. Western blotting for AMPKα (**A**) and lipolytic enzymes (ATGL and HSL) (**B**) in WAT. (**C**) Protein expression of AMPKα, ATGL, and HSL in 3T3-L1 adipocytes after differentiation for 8 days. Data are expressed as mean ± SD (*n* = 6). Values with different letters are significantly different; *p* < 0.05 (a > b > c > d).

**Figure 5 nutrients-10-01802-f005:**
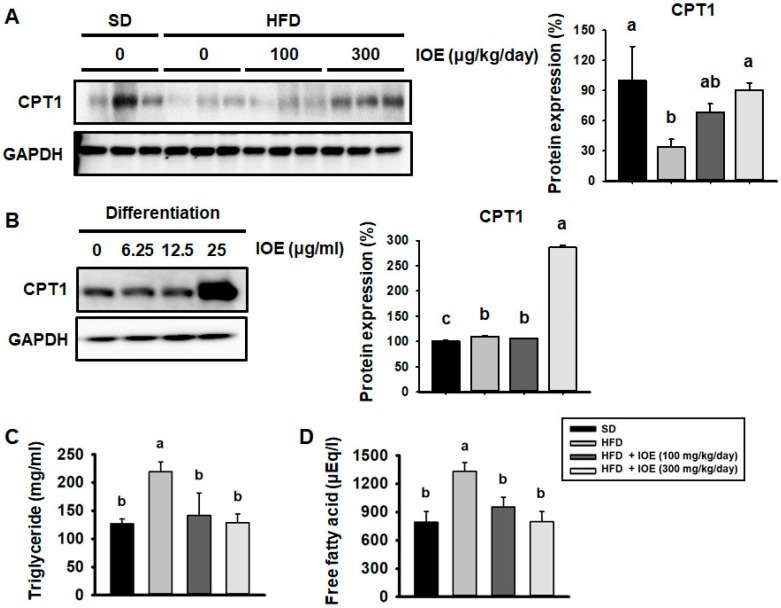
IOE increases the expression of carnitine palmitoyltransferase I (CPT1) in 3T3-L1 adipocytes and the WAT of HFD-induced obese mice, and reduces circulating TG and FFA concentrations. (**A**) Representative western blot showing CPT1 expression in WAT after 6 weeks of HFD feeding ± IOE administration. (**B**) Western blotting for CPT1 level in 3T3-L1 cells after differentiation ± IOE administration for 8 days. (**C**,**D**) Serum levels of TG and FFA were measured after 6 weeks of HFD feeding ± IOE administration. Data are expressed as mean ± S.D (*n* = 6). Values with different letters are significantly different; *p* < 0.05 (a > b).

**Figure 6 nutrients-10-01802-f006:**
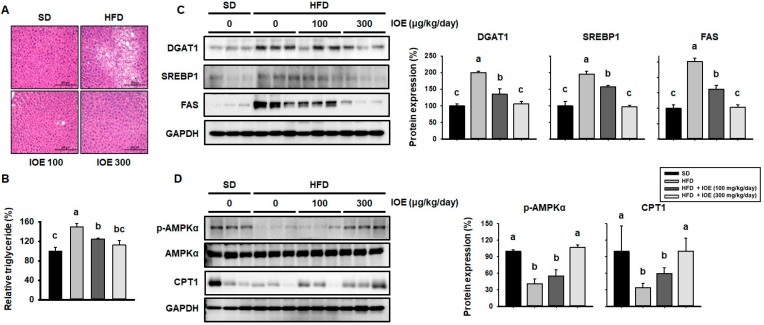
IOE treatment reduces hepatic lipid accumulation and regulates expression of proteins involved in lipogenesis and oxidation in HFD-induced obese mice. (**A**) Representative H&E-stained liver sections from mice fed a HFD ± IOE for 6 weeks. (**B**) Hepatic TG content. Protein levels of lipogenic genes (DGAT1, SREBP1, and FAS) (**C**) and β-oxidation genes (AMPKα and CPT1) (**D**) after HFD feeding ± IOE treatment for 6 weeks. Data are expressed as mean ± S.D (*n* = 6). Values with different letters are significantly different; *p* < 0.05 (a > b > c). DGAT1, diglyceride acyltransferase 1. SREBP1, sterol regulatory element-binding protein 1. FAS, fatty acid synthase. GAPDH, glyceraldehyde-3-phosphate dehydrogenase.

**Table 1 nutrients-10-01802-t001:** Effect of supplementation with IOE on organ mass in mice that were HFD-fed for 6 weeks.

Variables/Groups	Organ Mass (g)			
	SD	HFD		
	IOE 0 *	IOE 0 *	IOE 100 *	IOE 300 *
**Liver**	1.59 ± 0.27	1.69 ± 0.44	1.57 ± 0.67	1.41 ± 0.2
**Heart**	0.20 ± 0.03	0.22 ± 0.04	0.26 ± 0.08	0.24 ± 0.06
**Lung**	0.28 ± 0.02	0.35 ± 0.22	0.33 ± 0.08	0.25 ± 0.03
**Kidney**	0.68 ± 0.12	0.62 ± 0.12	0.57 ± 0.07	0.65 ± 0.14
**Spleen**	0.12 ± 0.03	0.14 ± 0.03	0.13 ± 0.03	0.11 ± 0.02

* (mg/kg/day). Data are expressed as mean ± standard deviations (S.D) (*n* = 6). HFD, high fat diet. IOE, *Ishige okamurae* extract.

**Table 2 nutrients-10-01802-t002:** Effect of IOE administration on blood parameters in mice that were HFD-fed for 6 weeks.

Group	Blood Parameter (mg/dL)			
	SD	HFD		
	IOE 0 *	IOE 0 *	IOE 100 *	IOE 300 *
**Total cholesterol**	120.8 ± 24.6 ^b^	171 ± 19.2 ^a^	123.8 ± 14.5 ^b^	119.6 ± 23.3 ^b^
**LDL-cholesterol**	9.4 ± 1.7 ^b^	27.4 ± 7.9 ^a^	12.8 ± 2.2 ^b^	11 ± 3 ^b^
**HDL-cholesterol**	121.4 ± 2.5 ^a^	89.2 ± 8.8 ^b^	114.8 ± 8 ^a^	113 ± 8.8 ^a^
**ALT**	30.8 ± 3.1 ^b^	54.8 ± 4 ^a^	34.2 ± 5.2 ^b^	31.4 ± 3.1 ^b^
**AST**	56.2 ± 10.6 ^b^	83.8 ± 7.2 ^a^	67.8 ± 6.1 ^b^	58.8 ± 7.4 ^b^
**LDH**	224 ± 26.6 ^b^	401.4 ± 123.1 ^a^	326.3 ± 32.8 ^b^	235.6 ± 39.4 ^b^

* (mg/kg/day). Data are expressed as mean ± S.D (*n* = 6). Values with different letters are significantly different, *p* < 0.05 (a > b). LDL, low-density lipoprotein. HDL, high-density lipoprotein. ALT, alanine transaminase. AST, aspartate transaminase. LDH, lactate dehydrogenase.

**Table 3 nutrients-10-01802-t003:** Effect of treatment with IOE on fasting blood glucose in mice that were HFD-fed for 6 weeks.

Group	Fasting Blood Glucose (mg/dL)						
	0 Week	1 Week	2 Weeks	3 Weeks	4 Weeks	5 Weeks	6 Weeks
**SD**	93.2 ± 13.5	87.4 ± 12.6 ^b^	88.6 ± 5.8 ^b^	104.6 ± 3.6 ^b^	90.0 ± 8.8 ^c^	91.8 ± 11.0 ^c^	97.2 ± 10.2 ^b^
**HFD**	103.4 ± 8.2	131.8 ± 20.3 ^a^	131.2 ± 17.6 ^a^	142.4 ± 19.1 ^a^	148.0 ± 16.0 ^a^	141.6 ± 7.6 ^a^	156.0 ± 27.2 ^a^
**HFD+** **IOE 100 ***	100.6 ± 21.1	106.8 ± 13.6 ^b^	110.2 ± 17.3 ^a,b^	119.2 ± 5.3 ^b^	118.4 ± 5.5 ^b^	118.4 ± 11.2 ^b^	103.2 ± 6.1 ^b^
**HFD+** **IOE 300 ***	97.0 ± 10.8	106.6 ± 7.7 ^b^	107.4 ± 14.3 ^a,b^	112.4 ± 18.7 ^b^	105.0 ± 13.3 ^b,c^	119.2 ± 8.6 ^b^	102.4 ± 14.1 ^b^

* (mg/kg/day). Data are expressed as mean ± S.D (*n* = 6). Values with different letters are significantly different, *p* < 0.05 (a > b > c).

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
