# Peer review of "Ishige okamurae Extract Suppresses Obesity and Hepatic Steatosis in High Fat Diet-Induced Obese Mice"

_nutrients, 2018, doi:10.3390/nu10111802_

Reviewer 1 Report

This is a largely descriptive study that shows changes in the expression of adipogenic transcription factors PPAR-gamma and C/EBP-alpha that correspond with beneficial changes in lipid metabolism and glucose homeostasis.  Both in vitro (3T3-L1 adipocytes) and in vivo (high fat fed mice) models are used.  The methodologies used are straightforward and appropriate for this study.  The conclusions reached are generally well supported by the data presented (with a few minor exceptions as noted below).  Given the high prevalence of obesity and continued need to develop effective strategies to curb this epidemic, the contributions made by this manuscript are helpful. In particular, novel insight into molecular changes that occur with ingestion of Ishige okamurae extract can guide efforts to exploit this effect.  In this regard, the paper would be significantly strengthened by efforts to elucidate the specific component(s) in the extract that drive these changes, or at least include a greater discussion of potentially important extract components. 

 1.      It would be helpful to correlate the dose of IOE used in the in vitro experiments with those that are achievable in vivo following extract consumption.

2.      The authors repeatedly claim that IOE “regulates” lipid synthesis and beta oxidation.  This is too strong of a conclusion.  The data actually shows that the extract influences these pathways, but no evidence is shown that this is the result of regulatory modulation. 

3.      In figure 2, the information in panel b is included in panel a. 

4.      The statement that brown fat mass “tended to be higher” (line 192) is not appropriate and implies a bias in viewing the data.  There is no statistical significance between treated and untreated groups.  Using the same criteria, one could say that food intake (as shown in panel c) tended to be lower.  This sentence should be removed or changed to indicate that there was no statistical difference between treated and untreated groups.

5.      In figure 4, it appears that the different bars in panel c represent different concentration of IOE rather than the groups shown in panels a & b.  This is unnecessarily confusing.  It would be helpful to designate this on the x-axes of the figures and use a single bar color.

6.      In line 265, it is incorrectly stated that “HFD-fed mice had lower TG content…” rather than higher TG content as shown in Fig 6b

7.      It is suggested that the authors test for changes in FGF-21 expression in response to IOE exposure in high fat fed mice (c.f. Clin Endocrinol 2013 Apr;78(4):489-96).

8.      The introduction and discussion both include basic descriptions of pathways relevant to fuel homeostasis.  This may be helpful to a reader not already acquainted with the field but could be substantially condensed, allowing for greater discussion of the specific influence of IOE components in the effects observed.

Author Response

Dear Chief in Editor and reviewers,

Thank you for considering our manuscript for publication in Nutrients. We have addressed the reviewer’s comments point-by-point and made the necessary changes to the manuscript.

We hope that the manuscript is now acceptable for publication in Nutrients.

 Sincerely,

Boo-Yong Lee

 # Reviewer 1

This is a largely descriptive study that shows changes in the expression of adipogenic transcription factors PPAR-gamma and C/EBP-alpha that correspond with beneficial changes in lipid metabolism and glucose homeostasis.  Both in vitro (3T3-L1 adipocytes) and in vivo (high fat fed mice) models are used.  The methodologies used are straightforward and appropriate for this study.  The conclusions reached are generally well supported by the data presented (with a few minor exceptions as noted below).  Given the high prevalence of obesity and continued need to develop effective strategies to curb this epidemic, the contributions made by this manuscript are helpful. In particular, novel insight into molecular changes that occur with ingestion of Ishige okamurae extract can guide efforts to exploit this effect.  In this regard, the paper would be significantly strengthened by efforts to elucidate the specific component(s) in the extract that drive these changes, or at least include a greater discussion of potentially important extract components.

 We are very pleasure to have been given the opportunity to revise our manuscript. We carefully revised our manuscript according to reviewer’s comments.

 1.      It would be helpful to correlate the dose of IOE used in the in vitro experiments with those that are achievable in vivo following extract consumption.

Response 1 → We conducted an XTT assay to determine the concentration of IOE in vitro. On the other hand, the method of determining the concentration in vivo has been referred to the reported paper. We set the human dosage of 1 g/day considering the formulation of health functional foods and the like. When making a healthy functional food, if a person consumes more than 1g per day, the dosage should not exceed 1g daily dosage. According to a well-known paper, it has been reported that we have to use the conversion factor to convert human dose to mouse dose [1,2]. We derived converting 1 g/day human (60kg) dose to 200 mg/kg/day mouse dose from reported paper. As a result, 200 mg/kg/day is the concentration between 100 and 300 mg/kg/day that we set.

 2.      The authors repeatedly claim that IOE “regulates” lipid synthesis and beta oxidation.  This is too strong of a conclusion.  The data actually shows that the extract influences these pathways, but no evidence is shown that this is the result of regulatory modulation.

Response 2 → We modified to avoid direct claim in abstract, introduction, discussion, and conclusion. (ex. ‘IOE regulates lipid synthesis and beta oxidation“-related factors”.’)

3.      In figure 2, the information in panel b is included in panel a.

Response 3 → We modified panel b to include in panel a.

4.      The statement that brown fat mass “tended to be higher” (line 192) is not appropriate and implies a bias in viewing the data. There is no statistical significance between treated and untreated groups.  Using the same criteria, one could say that food intake (as shown in panel c) tended to be lower.  This sentence should be removed or changed to indicate that there was no statistical difference between treated and untreated groups.

Response 4 → We modified it to another word as your kind comment.

(‘brown adipose tissue (BAT) mass tended to increase in IOE-treated than solely HFD-fed mice but there is no statistical significance between treated and untreated groups. As shown in Figure 2b, food intake was no statistical difference.’)

5.      In figure 4, it appears that the different bars in panel c represent different concentration of IOE rather than the groups shown in panels a & b.  This is unnecessarily confusing.  It would be helpful to designate this on the x-axes of the figures and use a single bar color.

Response 5 → We added figure description in figure 4c. We added the description below the graph as in Figure 1c and labeled it as 3T3-L1 cells.

6.      In line 265, it is incorrectly stated that “HFD-fed mice had lower TG content…” rather than higher TG content as shown in Fig 6b

Response 6 → We modified it to another word as your kind comment. (lower → higher)

7.      It is suggested that the authors test for changes in FGF-21 expression in response to IOE exposure in high fat fed mice (c.f. Clin Endocrinol 2013 Apr;78(4):489-96).

Response 7 → FGF-21 is mainly expressed in liver and adipose tissue. FGF-21 treatment suppresses body weight and blood glucose and enhances energy expenditure. Moreover, FGF-21 plays a key role in the treatment of type 2 diabetes. In addition to this article, we are currently conducting an experiment on glucose and energy metabolism by IOE in db/db mice. We will have to improve the quality of paper by observing changes in FGF-21 expression by treatment with IOE.

8.      The introduction and discussion both include basic descriptions of pathways relevant to fuel homeostasis. This may be helpful to a reader not already acquainted with the field but could be substantially condensed, allowing for greater discussion of the specific influence of IOE components in the effects observed.

Response 8 → We explained in more detail according to your kind comments in discussion part.

(‘IOE contained 4.2% of polyphenols and Ishphloroglucin A, an indicator component of IOE, was also composed of polyphenol. Polyphenol-rich compounds are well known to have anti-obesity and blood glucose control activity [3]. It has been reported that polyphenol-rich compound inhibits body weight, hyperglycemia, and hyperinsulinemia via enhancement of energy expenditure and fat oxidation [3,4]. Previous studies have shown that diphlorethohydroxycarmalol isolated from Ishige okamurae reduces blood sugar [5]. We anticipated that IOE might have anti-obesity properties. The results of this investigation demonstrate that IOE reduces adipose tissue mass, total body mass, and hepatic lipid content.’).                                                         

Reference

1.         Freireich, E.J.; Gehan, E.A.; Rall, D.P.; Schmidt, L.H.; Skipper, H.E. Quantitative comparison of toxicity of anticancer agents in mouse, rat, hamster, dog, monkey, and man. Cancer Chemother Rep 1966, 50, 219-244.

2.         Reagan-Shaw, S.; Nihal, M.; Ahmad, N. Dose translation from animal to human studies revisited. FASEB J 2008, 22, 659-661.

3.         Meydani, M.; Hasan, S.T. Dietary polyphenols and obesity. Nutrients 2010, 2, 737-751.

4.         Klaus, S.; Pültz, S.; Thöne-Reineke, C.; Wolfram, S. Epigallocatechin gallate attenuates diet-induced obesity in mice by decreasing energy absorption and increasing fat oxidation. International journal of obesity 2005, 29, 615.

5.         Min, K.H.; Kim, H.J.; Jeon, Y.J.; Han, J.S. Ishige okamurae ameliorates hyperglycemia and insulin resistance in c57bl/ksj-db/db mice. Diabetes Res Clin Pract 2011, 93, 70-76.

We hope that you will consider this manuscript for publication in Nutrients.

Reviewer 2 Report

This paper is looking at the influence of an extract in obesity and NASH, two very important conidtions/diseases in the world. The paper is well written with a sound study design. The discussion could have a better emphasis on the relevance of the extract as it doesn't really come through just now. I have noted some minor points below:

- Page 3:

    - line 91 "we got powder...": this should be rephrased.

    - line 112 the number of cells reads x104 instead of 104.

    - line 118 and 135 "CO2" should be CO2

- Page 4:

    - line 146/147, Cayman chemical is missing the location.

- Page 6:

    - Table 1: there is no stastistical significance so the "a" should really be removed as it doesn't add anything to the information provided. If you wish to keep it, then it should be noted in the legend.

Author Response

Dear Chief in Editor and reviewers,

Thank you for considering our manuscript for publication in Nutrients. We have addressed the reviewer’s comments point-by-point and made the necessary changes to the manuscript.

We hope that the manuscript is now acceptable for publication in Nutrients.

 Sincerely,

Boo-Yong Lee

# Reviewer 2

This paper is looking at the influence of an extract in obesity and NASH, two very important conidtions/diseases in the world. The paper is well written with a sound study design. The discussion could have a better emphasis on the relevance of the extract as it doesn't really come through just now. I have noted some minor points below:

We are very pleasure to have been given the opportunity to revise our manuscript. We carefully revised our manuscript according to reviewer’s comments. We added a more detailed description of the IOE according to your kind feedback in the discussion section.

- Page 3:

    - line 91 "we got powder...": this should be rephrased.

Response 1 → We modified it to another word as your kind comment. (we got powder it was powdered)

    - line 112 the number of cells reads x104 instead of 104.

Response 2 → We changed it. (104 104)

    - line 118 and 135 "CO2" should be CO2

Response 3 → We changed it. (CO2 CO2)

- Page 4:

    - line 146/147, Cayman chemical is missing the location.

Response 4 → We added it. (Cayman Chemical Company, Ann Arbor, MI, USA)

- Page 6:

    - Table 1: there is no statistical significance so the "a" should really be removed as it doesn't add anything to the information provided. If you wish to keep it, then it should be noted in the legend.

Response 5 → We removed it.

We hope that you will consider this manuscript for publication in Nutrients
